# Refining the Intraoperative Identification of Suspected High-Grade Glioma Using a Surgical Fluorescence Biomarker: GALA BIDD Study Report

**DOI:** 10.3390/jpm13030514

**Published:** 2023-03-13

**Authors:** Colin Watts, Alimu Dayimu, Tomasz Matys, Keyoumars Ashkan, Stephen Price, Michael D. Jenkinson, Gail Doughton, Claire Mather, Gemma Young, Wendi Qian, Kathreena M. Kurian

**Affiliations:** 1Academic Department of Neurosurgery Institute of Cancer and Genomic Sciences, University of Birmingham, Birmingham B15 2TT, UK; 2Clinical Trials Unit, Department of Oncology, University of Cambridge, Cambridge CB2 0QQ, UK; 3Department of Radiology, University of Cambridge, and Cambridge University Hospitals NHS Foundation Trust, Cambridge Biomedical Campus, Cambridge CB2 0QQ, UK; 4Department of Neurosurgery, King’s College Hospital, London SE5 9RS, UK; 5Academic Neurosurgery Unit, Department of Clinical Neurosciences, University of Cambridge, Cambridge CB2 0QQ, UK; 6Department of Neurosurgery, Walton Centre, University of Liverpool, Liverpool L9 7LJ, UK; 7Cambridge Clinical Trials Unit—Cancer Theme (CCTU-CT), Cambridge CB2 0QQ, UK; 8Brain Tumour Research Centre, University of Bristol Medical School & North Bristol Trust, Bristol BS10 5NB, UK

**Keywords:** 5ALA, intraoperative biomarker, fluorescence

## Abstract

Background. Improving intraoperative accuracy with a validated surgical biomarker is important because identifying high-grade areas within a glioma will aid neurosurgical decision-making and sampling. Methods. We designed a multicentre, prospective surgical cohort study (GALA-BIDD) to validate the presence of visible fluorescence as a pragmatic intraoperative surgical biomarker of suspected high-grade disease within a tumour mass in patients undergoing 5-aminolevulinic acid (5-ALA) fluorescence-guided cytoreductive surgery. Results. A total of 106 patients with a suspected high-grade glioma or malignant transformation of a low-grade glioma were enrolled. Among the 99 patients who received 5-ALA, 89 patients were eligible to assess the correlation of fluorescence with diagnosis as per protocol. Of these 89, 81 patients had visible fluorescence at surgery, and 8 patients had no fluorescence. A total of 80 out of 81 fluorescent patients were diagnosed as high-grade gliomas on postoperative central review with 1 low-grade glioma case. Among the eight patients given 5-ALA who did not show any visible fluorescence, none were high-grade gliomas, and all were low-grade gliomas. Of the seven patients suspected radiologically of malignant transformation of low-grade gliomas and with visible fluorescence at surgery, six were diagnosed with high-grade gliomas, and one had no tissue collected. Conclusion. In patients where there is clinical suspicion, visible 5-ALA fluorescence has clinical utility as an intraoperative surgical biomarker of high-grade gliomas and can aid surgical decision-making and sampling. Further studies assessing the use of 5-ALA to assess malignant transformation in all diffuse gliomas may be valuable.

## 1. Introduction

Although brain tumours in adults account for less than 2% of all primary tumours, the average years of life lost per patient is the most of any cancer with over 20 years of life lost [1]. Recent trends in survival between 2000-04 and 2010-14 show limited improvement (range 3–10%), reflecting the limited therapeutic options [2,3]. 

The most aggressive adult brain high-grade glioma is the glioblastoma (GB) [4], which accounts for over 60% of brain tumours and has a median life expectancy in optimally managed patients of only 13–16 months with less than 25% surviving 24 months [5,6]. The current clinical management of patients diagnosed with a GB or another high-grade glioma involves a combination of surgery, radiotherapy and chemotherapy [3]. The objective of surgery is to obtain tissue to establish a histopathological diagnosis and to remove as much tumour as possible to improve oncological outcomes, i.e., cytoreduction while preserving eloquent pathways to maintain health-related quality of life and preserve neurocognitive outcomes [7]. Good cytoreduction is associated with improved survival, but the quality of the data, based on GRADE criteria, is moderate to low [8,9]. Surgical removal of high-grade gliomas can be confounded by proximity to important neurovascular structures and the eloquent brain and by the difficulty visualizing the infiltrative tumour margin, resulting in tumour tissue being inadvertently left behind [7,10]. Surgical adjuncts to help the surgeon identify tumours during surgery include intraoperative MRI [11], ultrasound [12] and fluorescence [13]. However, the evidence is of limited quality due to a lack of prospective trial data [14]. 

A prospective study of 5-aminolevulinic acid (5-ALA) to aid in visualization of malignant gliomas during surgery has provided the most robust evidence of improved cytoreduction [10,15], which in turn can improve clinical outcomes [8,9]. In addition, 5-ALA is an endogenous intermediate of the porphyrin biosynthesis pathway. It acts as a prodrug that is metabolized intracellularly to form the fluorescent molecule protoporphyrin IX (PPIX). The exogenous application of 5-ALA leads to a highly selective accumulation of PPIX in tumour cells [16,17,18,19], and biopsy studies have confirmed that a normal brain does not fluoresce [20].

Absorption of light excites the PPIX molecules from their ground state. This effect is most efficient at wavelengths of approximately 400 nm (blue light). Decay from the excited state is accompanied by the emission of red light (λmax = 635 nm) that can be visualised intraoperatively as pink fluorescence using specially modified operative microscopes.

Tumour visualization technologies, when used appropriately, facilitate the collection of diagnostic tumour samples. Accurate diagnosis is essential to optimize treatment and aid prognostic counselling and postoperative management [21,22,23]. Improving intraoperative diagnostic accuracy is important because knowing that a tumour is high-grade will aid surgical decision-making and sampling [13,24]. Nests of high-grade glioma (HGG) can be “hidden” in some tumours that appear less aggressive and because the nests cannot be seen they may be missed at surgery [25,26,27]. 

For these reasons, it is increasingly important to provide the surgeon with as much information as possible during surgery to allow the procedure to be tailored to optimize benefit for each individual patient. The surgical benefit would be enhanced through improved safety and efficacy of the operation while providing sufficient material for an accurate diagnosis. To address this, we designed a multicentre prospective surgical study to validate the presence of visible fluorescence as a pragmatic intraoperative surgical biomarker of high-grade disease within a tumour mass. 

## 2. Methods

GALA-BIDD was a multicentre prospective diagnostic surgical intraoperative biomarker study. The primary objective of this study was to validate the presence of visible 5-ALA fluorescence as a pragmatic intraoperative diagnostic surgical biomarker of high-grade disease within a tumour mass. The secondary objective was to describe the patient characteristics and disease histology for eligible patients who had no visible fluorescence at surgery. 

The study schema is outlined in Figure 1. 

### 2.1. Patient Eligibility and Registration

Patients were eligible if they were aged 16 years or older, had a WHO performance status (WHO PS) of 0 or 1, had radiological suspicion of an intrinsic glioma WHO grade II that was transforming to a higher grade or a suspected WHO grade III/IV tumour and had been reviewed by a specialist neuro-oncology multidisciplinary team (MDT) in which fluorescence guided cytoreductive surgery was judged to be appropriate. 

Patients were excluded if there was a previous history of other malignancy (except for adequately treated basal/squamous cell carcinoma or carcinoma in situ) within 5 years, for patients with previous brain surgery (including biopsy) or cranial radiotherapy and for patients being treated with anticancer therapy (except steroids). In addition, patients who had known or suspected allergies to 5-ALA or porphyrins or acute or chronic types of porphyria; had symptomatic acute liver disease, platelets <100 × 10^9^/L or were pregnant or lactating were excluded from this study because 5ALA is contraindicated in these groups. 

Eligible patients were recruited prospectively from 3 UK neurosurgical centres and centrally registered at the Cambridge Clinical Trials Unit prior to the collection of patient data or samples. Screening/baseline assessments were conducted up to 4 weeks prior to surgery. All patients provided written informed consent before enrolment. 

### 2.2. Magnetic Resonance Imaging (MRI) Tumour Assessment

Patients underwent routine clinical MRI scanning according to local standard of care protocols, including contrast-enhanced MRI prior to the surgery and within 72 h of surgery. Guidance for MR imaging was that MRI machines should be 3 Tesla (≥1.5 Tesla is acceptable) where possible, and there should not have been any switching between magnets of different field strengths, parameters, and modalities, for MRI images should remain consistent for all the study’s MR images. The patient’s postoperative MRI report was collected to assess the extent of resection. Both baseline and postoperative MRIs were reviewed centrally by an independent neuro-radiographer for all patients. Low-grade glioma cases with suspected high-grade transformation were also independently assessed by central neuroradiological review/opinion (TM).

### 2.3. Neurosurgery and Tumour Sample Collection

Patients recruited into this study underwent cytoreductive surgery according to local standard of care. The use of specific surgical adjuncts to aid resection other than 5-ALA was left to the individual surgeon’s preference. Then, 5-ALA was administered orally 3–4 h before surgery as per recommended guidelines at a dose of 20 mg/kg.

The tissue samples were collected during surgery and sent for histopathological analysis including both intraoperative (hotline) and postoperative (offline) specimens. Samples were collected and processed in a standardized manner (see Figure 2). 

Patients with visible fluorescence during surgery had an initial sample of fluorescing tumour collected and bisected by the operating surgeon into two equal pieces with both pieces being sent to pathology at the same time during surgery to avoid spatial sampling bias. One piece was used for perioperative analysis and one piece for postoperative analysis as follows:One piece was evaluated by intraoperative histopathological analysis (hotline specimen) as per standard local procedures (based on a smear AND a frozen section) to provide intraoperative diagnosis.The other piece was analysed postoperatively (offline processing including normal fixation AND paraffin embedding) as per standard local procedures. The postoperative histopathological grade of this individual piece was reported locally, as well as contributing to the overall histopathological diagnosis of the patient’s glioma.

Remaining tumour samples collected during surgery were sent for postoperative offline processing (including normal fixation and paraffin embedding) as per standard local procedures to establish a definitive histopathological diagnosis.

Tumour samples from patients who underwent 5-ALA treatment but where visible fluorescence was NOT present during surgery were collected using the same standardized approach as those with visible fluorescence. 

### 2.4. Local and Independent Central Molecular Tumour Assessment 

After surgery, details were collected on each patient’s local postoperative histopathological diagnosis and an anonymised copy of the associated histopathology report. Standard-of-care molecular diagnostic tests IDH-1 mutation, MGMT methylation status and 1p19q deletion performed locally were also collected. 

Diagnostic histopathology reports together with the full molecular diagnostic tests report were reviewed centrally by a neuropathologist blind to the local intraoperative diagnosis. When full molecular diagnostic tests were not performed as standard of care, these tests were carried out retrospectively by the central pathology reviewer using local paraffin-embedded samples. The final overall postoperative diagnosis was based on WHO 2016 CNS criteria [28] where possible through independent central review. Since this study was completed, the WHO 2021 classification has been published in which the grading incorporates molecular and histological parameters [29]. For the purposes of our study, the main histological grading criteria which are used at the time of intraoperative diagnosis have not changed in the 2021 classification, and therefore we have opted to use the classification given at the time where possible, incorporating the molecular information that was available at the time of this study.

### 2.5. Clinical Management

Clinical treatment, including radiotherapy and chemotherapy, was managed according to the local site standard procedures in line with current standard of care. The overall survival (OS) was collected until 6 months after the last patient enrolled. 

### 2.6. Endpoints and Statistical Analysis

The primary endpoint of this study was the percentage of patients with visible fluorescence who had confirmed high-grade glioma that was defined as a WHO 2021 CNS grade 3 or 4 by central review. 

With a 1% significance level (one-sided) and 95% power, 80 patients with visible fluorescence were required to detect the percentage of fluorescence patients with confirmed high-grade disease being <90% vs. >99%. To allow for a 20% noncompliance rate or no visible fluorescence, it was planned to recruit 100 patients. 

The statistical analyses were mainly descriptive. Primary endpoint was based on all eligible patients who had surgery with 5-ALA and central overall postoperative diagnosis. The Clopper–Person method was used for the 99% confidence interval of the primary endpoint. Results were presented as the mean ± standard deviation, median (range) for continuous variables or percentage together with the confidence interval for category variables wherever appropriate. Survival outcome was presented using the Kaplan–Meier approach. Statistical analyses were performed using the Statistical Analysis System (SAS) software package (version 9.4; SAS Institute Inc., Cary, NC, USA).

### 2.7. Ethical Approval and Sponsorship

This study was funded by CRUK (ref C9221/A16700). Cancer Research UK had no role in study design, data collection, data analysis, data interpretation or writing of the report. The study sponsor, the University of Cambridge, was responsible for enrolment, data collection, entry, and validation; monitoring procedures; organization of central review; liaison with investigators; data analyses and writing of the report. The sponsor had no role in study design or data interpretation. This study was approved by South Central—Hampshire B Research Ethics Committee (14/SC/1126) and was conducted in accordance with the Declaration of Helsinki and Good Clinical Practice.

## 3. Results

### 3.1. Baseline Data

Between February 2015 and March 2017, a total of 106 patients were enrolled from 3 sites in the UK (Figure 3). The baseline characteristics (*n* = 106) are summarized in Table 1 and Appendix A. The median age was 59 (range: 23–77) years with 59% male patients. All patients were WHO Performance Status (PS) 0 or 1 with 51.4% patients PS 0 and 48.6% PS 1. A total of 5 (4.7%) patients had 2 tumour locations, and 6 (5.4%) were multifocal. Most of the tumours were in the frontal (40, 36.0%) and temporal (40, 36.0%) lobes and equally distributed between the left (50.0%) and right (50.0%) hemispheres. 

Preoperatively, 20/106 (18.9%) patients were described radiologically as suspicious of WHO grade 2 tumours possibly transforming to a higher grade and 84 (79.2%) as probably WHO grade 4. 

Five registered patients were ineligible and were withdrawn from this study prior to surgery (Figure 3): MRI was not performed prior to MDT for one patient; one patient was classified as WHO grade II upon MRI; two patients did not have liver function tests prior to registration, and one had previous cranial radiotherapy prior to resection.

One hundred and one patients underwent surgical resection (Figure 3). Two patients failed to receive 5-ALA due to an administration error. Therefore, 5-ALA was administered to 99/101 (98.0%) patients prior to surgery with a median dose of 1500 mg (range: 960–2200). The median time between the 5-ALA administration and the start of the surgery was 4 h (range: 1.2–8.6) with a time of less than 3 h for 7 patients (all these 7 patients had a final local histopathological diagnosis of high-grade disease). 

As summarized in Table 1, 85 of 99 patients (85.9%) who received 5-ALA had visible fluorescence during surgery, 14/99 had no visible fluorescence at surgery and 75/99 patients (75.8%) were deemed to have a complete neurosurgical resection based on the observation that there was no measurable contrast-enhancing disease on a postop MR obtained within 72 h of surgery. A total of 30 out of 63 patients (47.6%) did not achieve a complete resection of enhancing tumour (CRET), and only 20 (66.7%) of those patients had more than 90% of enhancing tumour resected.

Of the 99 patients given 5-ALA prior to surgery who were, therefore, eligible for diagnostic assessment, 10 patients were withdrawn. Five patients had no research tissue collected as per protocol (three with visible fluorescence and two without); four patients were not gliomas: two meningiomas (one with visible fluorescence), one ganglioglioma, one nonglial tumour and one patient was diagnosed as non-neoplastic tissue (without visible fluorescence) (Figure 3). A total of 89 patients were, therefore, eligible to assess the correlation of fluorescence with the WHO 2016 diagnosis as per protocol.

### 3.2. Correlation of Fluorescence with Molecular Pathology

A total of 89 patients were suitable for full analysis as per protocol (Appendix A). Of these, 81 of 89 patients (91%) showed visible fluorescence at surgery and 8 patients did not. A total of 80 out of 81 fluorescent patients (98.8%) were diagnosed with high-grade glioma upon postoperative central review. 

Histological analysis of the 81 fluorescent tumours reported that 75 (92.6%) were IDHwt WHO grade 4 glioblastoma, 1/81 (1.2%) was IDHmut WHO grade 4 astrocytoma, 2/81 (2.5%) were WHO grade 3 astrocytoma NOS, 2/81 (2.5%) were oligodendroglioma (1 WHO grade 2 and 1 WHO grade 3) and one case was missing. The standard-of-care molecular analysis reported that MGMT promotor methylation status was available for 78/81 (96.3%) fluorescent tumours. A total of 31/81 were methylated (29 WHO grade 4 GB and 2 WHO grade 3 astrocytoma NOS), and 47/81 were unmethylated, 46 WHO grade 4 GB together with 1 WHO grade 3 astrocytoma NOS. Additionally, 2 patients with visible fluorescence were 1p,19q codeleted (Appendix A), 1 patient was a WHO grade 2 oligodendroglioma and 1 patient was a WHO grade 3 oligodendroglioma.

Among the eight patients given 5-ALA who did not show any visible fluorescence, none were high-grade glioma (Appendix A). The IDH mutation status was available for 7/8 (87.5%). All 5 IDH-mutated patients were WHO grade 2 tumours. The methylation status was available for 3/8 patients in this group: 2 were methylated and 1 was unmethylated. The 1p19q codeletion status was available for 5/8 (62.5%) of these patients. A total of 2 were 1p19q intact WHO grade 2 astrocytoma NOS, and 3 were 1p19q codeleted WHO grade 2 oligodendroglioma.

### 3.3. Perioperative Reporting

The relationship between preoperative radiological interpretation, intraoperative and postoperative histopathological reporting of low or high-grade disease and the presence or absence of visible fluorescence at surgery is summarized in Table 2. 

Among the 81/99 patients with probable HGG on local preoperative imaging, the intraoperative analysis was unable to distinguish high- from low-grade gliomas in 16 (7 unable to report, 7 “other” and 2 missing) (17.3%) patients. There were 18/99 patients with preoperative suspicion of malignant transformation of LGG on local imaging. In this cohort, the intraoperative analysis was unable to report either high- or low-grade disease in 7/18 (39%) (2 other, 4 unable to report and 1 missing) patients. Visible fluorescence was reported in 7/18 (39%) patients at the surgery, while 11/18 patients (61%) had no visible fluorescence. Of the seven patients suspected radiologically of malignant transformation of LGG and with visible fluorescence at surgery, six were diagnosed with high-grade glioma, and one had no tissue collected.

Among the 99 patients who received 5-ALA, a perioperative diagnosis could not be made in 23/99 (23.2%). In 11 patients, the intraoperative diagnosis was recorded as “unable to report”, in 9 as “other” and in 3 as “missing data”.

Among the 89 patients eligible for full analysis as per protocol, there was an agreement between local and central histopathological analysis in 85/89 (95.5%) patients. However, the intraoperative diagnosis was consistent with the central histopathological report in only 73/89 (82.0%) patients. Eight patients were low-grade, and sixty-five patients were high-grade. The remaining 16/89 (18%) patients were classified as “other” or “unable to report” at perioperative diagnosis and demonstrated visible fluorescence at the surgery. Of these 16 fluorescent patients, 1 was a 1p19q codeleted WHO grade 2 oligodendroglioma, and 15 were diagnosed as WHO grade 4 GB. 

The correlation between the final postoperative classification and the presence or absence of visible fluorescence is summarized in Table 3. A total of 89 patients were suitable for full analysis as per protocol. Of these, 81 of 89 patients (91%) showed visible fluorescence at surgery, and 8 patients did not. A total of 80 out of 81 fluorescent patients (98.8%) were diagnosed as high-grade gliomas upon postoperative central review. 

In the context of clinical suspicion of a high-grade disease or malignant transformation of low-grade disease in this study, the sensitivity and specificity that visible fluorescence during surgery represents high-grade disease was 100% and 88.9% (99% CI 61.9% to 100%), respectively. Corresponding positive predictive values and negative predictive values were 98.8% (99% CI 95.6% to 100%) and 100%, respectively.

Based on these prospective data, if there is clinical suspicion based on preoperative clinical and imaging data, the sensitivity that 5-ALA truly identifies high-grade glioma through visible fluorescence during surgery is 100% (*n* = 81 patients), while the specificity that 5-ALA truly identifies the absence of high-grade disease through a lack of visible fluorescence of the tumour mass during surgery is 88.9% (*n* = 8 patients).

Where preoperative clinical suspicion is present, if the operating surgeon sees visible 5-ALA fluorescence in the tumour mass at surgery, the positive predictive value that this represents high-grade glioma is 98.8%, but if the surgeon cannot see visible 5-ALA fluorescence in the tumour mass, then the negative predictive value that this represents an absence of high-grade glioma is 100%.

### 3.4. Survival Outcomes

The estimated median OS was 18.1 (95% CI = 13.3–23.4) months for high-grade patients, and no death was observed in patients without visible fluorescence (Figure 4). Stratification by IDH-1 status showed the median OS to be 16.3 (95%CI = 12.9–20.4) months for patients with no IDH-1 mutation (*n* = 78), whereas the median OS was not reached for IDH-1 mutated patients (*n* = 8). Stratification by MGMT promotor methylation status showed a median OS of 12.9 months (95% CI = 11.8–18.6) in unmethylated patients compared to a median OS of 21.2 months for methylated patients (95% CI = 13.7–not reached).

## 4. Discussion

We report a prospective study to validate the presence of visible fluorescence as a pragmatic intraoperative diagnostic surgical biomarker of high-grade disease within a tumour mass. The utility of visible fluorescence was evaluated in the context of routine clinical standard of care in a specialist care pathway beginning with a specialist multidisciplinary team (SMDT) review [30,31]. Patients were recruited based on clinical suspicion of a high-grade glioma or malignant transformation of a low-grade glioma. 

Our data show that if there is preoperative clinical suspicion based on clinical and imaging data, the sensitivity that 5-ALA truly identifies high-grade glioma through visible fluorescence during surgery is 100% (*n* = 81 patients), while the specificity that 5-ALA truly identifies the absence of high-grade disease through a lack of visible fluorescence of the tumour mass during surgery is 88.9% (*n* = 8 patients). If the operating surgeon sees visible 5-ALA fluorescence in the tumour mass at surgery, the positive predictive value that this represents a high-grade glioma is 98.8%, but if the surgeon cannot see visible 5-ALA fluorescence in the tumour mass, then the negative predictive value that this represents an absence of high-grade glioma is 100%. It is essential to recognize that in this study, the absence of 5-ALA does not exclude low-grade glioma. The 5-ALA biomarker was not used in isolation but as part of a holistic, specialized approach to the surgical management of high-grade brain cancer. Clinical suspicion can be based on radiological appearances, which may be obvious appearances of a GB (or WHO grade 4 IDH mutant astrocytoma) and described as “probable high-grade disease on pre-operative imaging” in this study. Alternatively, the imaging may be more subtle, such as a small focus of weak enhancement in an otherwise multienhancing lesion. These patients were described as having “suspected transformation to high-grade tumour on preoperative imaging”. Other factors include neurological symptoms, new onset seizures or changes in seizure activity or functional changes. Age is also an important factor [23,32]. 

When integrated into specialist care, pathway 5-ALA-based fluorescence can act as an intraoperative surgical biomarker of high-grade disease. The development pathway shares similarities with biomarker development roadmaps for both imaging data and biospecimens [33,34]. Our data presented here are the first prospective study to implement a surgical biomarker approach that incorporates a roadmap strategy (see Appendix A) with recognized standards for reporting diagnostic accuracy [35]. We show that visible fluorescence is highly accurate in predicting high-grade disease and highlights a previously under-recognized utility in the context of suspected malignant transformation of low-grade gliomas in preoperative challenging cases. Previous studies have demonstrated that 5-ALA-induced fluorescence is capable of visualizing anaplastic foci during surgery [25,26,36,37,38]. More recently, ex vivo studies have shown 5-ALA “hotspots” adjacent to blood vessels in low-grade glioma [39] and heme biosynthesis transcriptional profiling reveals a strong association between increasing mRNA expression and reduced survival [40]. However, mechanistic detail remains to be determined, and further prospective research is required.

Fluorescence-guided surgery using alternatives to 5-ALA has also been investigated. The most widespread alternative is sodium fluoresceine (SF). Recent studies suggest that this could be a viable alternative [41,42]. However, the level of evidence is weaker than that for 5-ALA. An alternative approach is exploring the potential of an agent used in photodynamic therapy, talaporfin sodium, as a fluorophore for intraoperative diagnosis [43]. Initial promising data require further validation and wider evaluation.

An important element in developing an intraoperative biomarker is to determine superiority (or arguably multi-inferiority) to standard intraoperative histopathological analysis. Our data showed that in patients with a probable high-grade disease on preoperative imaging, the perioperative analysis was unable to distinguish high- from low-grade disease in 17.3% of cases. The subsequent postoperative analysis confirmed that 95% of those with visible fluorescence were high-grade with 1 fluorescent low-grade (in a 77-year-old man) and 3 missing. There were three patients thought likely to be high-grade that turned out to be low-grade in two cases, and one was not a glioma. None of these tumours displayed visible fluorescence.

Similarly, intraoperative analysis was unable to distinguish high- or low-grade disease in 39% of patients suspected of malignant transformation based on preoperative imaging. Seven of these patients displayed visible fluorescence, and six of those seven patients were subsequently confirmed to be high-grade tumours. The remaining patient had no tissue collected. Among the multifluorescent patients, six were confirmed low-grade, and five could not be reported.

These data confirm the value of visible fluorescence as a biomarker of high-grade disease in mitigating the need for further biopsies when there is uncertainty about intraoperative analysis. Indeed, it could be argued that there is no need for an intraoperative analysis in fluorescent tumours [44], thereby improving workflow and reducing operating times with associated cost-efficiency savings. However, our observations are based on a crude, pragmatic stratification of high- vs. low-grade disease. Improvement in rapid molecular characterization particularly with reference to suspected low-grade pathologies in a timeframe that can influence surgical decision-making can be integrated into clinical workflows [45,46,47]. This would provide real-time molecular data of prognostic significance [22,48] that would allow for refinement of the surgical procedure to further mitigate risk and improve patient quality of life while accelerating diagnosis.

Local real-world molecular classification of our cases demonstrated the expected survival stratification of high-grade and low-grade glioma cases as predicted by 5ALA within the timeframe of this study. Our survival data are consistent with modern data reported in other trials [6,23,49] and reflect the overwhelming prevalence of IDHwt GB in the fluorescent population. 

Quantification of the clinical utility of visible fluorescence has been reported using standard metrics. Additional quality assurance measures were taken to mitigate bias and confounding factors [35]. These included taking samples within defined tumour boundaries relative to the margin, standardizing how the samples were taken, quantifying the time from 5-ALA administration to the start of surgery, defining fluorescence in binary terms (visible or multi-visible) and appropriate statistical design [35,50]. Independent study management through a clinical trial unit also allowed us to identify when these standards were not followed or when tissue samples were not taken. The use of centralized reporting of imaging and histopathological reporting was also integral to our overall quality control strategy to deliver robust evidence in the surgical setting.

These efforts allowed us to describe, in a real-world scenario, the reasons why a high proportion of initially enrolled patients were subsequently withdrawn: 17/106 (16%). Of these, 5/106 (4.7%) were deemed ineligible early after initial enrolment. In 2/106 (1.8%), 5ALA was not administered, and 10/106 were withdrawn postoperatively: In 5 patients, tissue was not collected, and 5 patients were deemed ineligible postoperatively after central neuropathology review. This information highlights the need for additional research nurse/technical support to assist with sample collection and emphasizes the value of good-quality sample collection.

In summary, we demonstrate for the first time in a prospective multi-center cohort study that 5ALA-based fluorescence as a pragmatic surgical biomarker is sensitive and specific for high-grade disease within an intrinsic glioma mass. We also demonstrate that it can enhance local intraoperative neuropathology diagnosis. Our data indicate that 5ALA may help as an adjunct diagnostic surgical biomarker for challenging low-grade glioma cases with clinical suspicion of transformation to higher-grade disease, but further research is required.

## Figures and Tables

**Figure 1 jpm-13-00514-f001:**
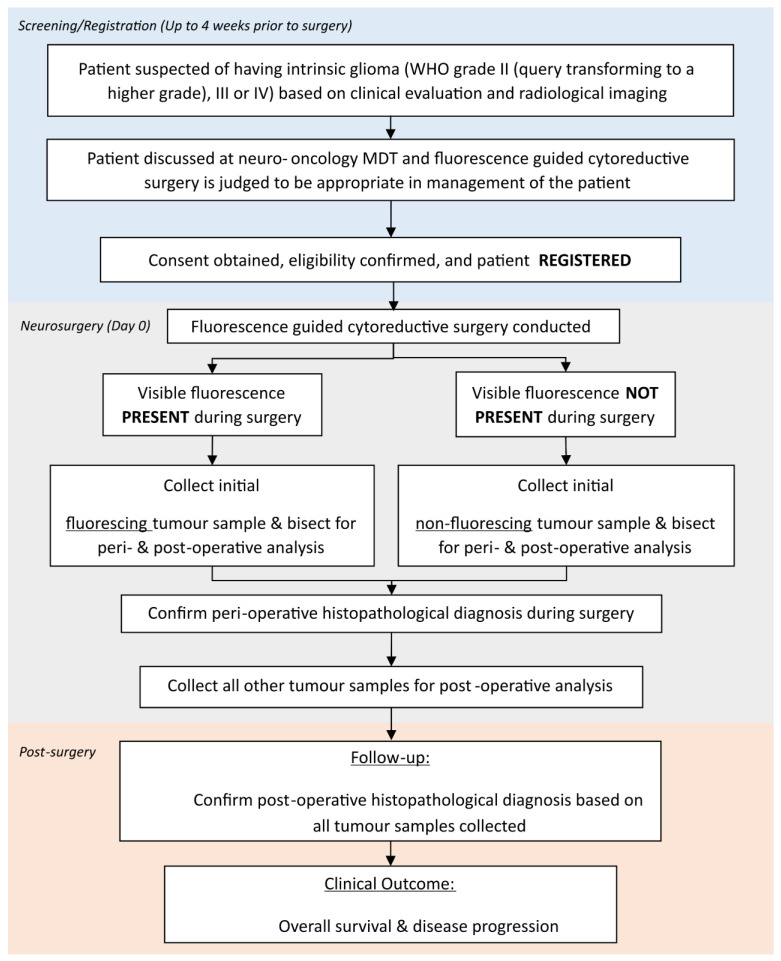
Study Schema.

**Figure 2 jpm-13-00514-f002:**
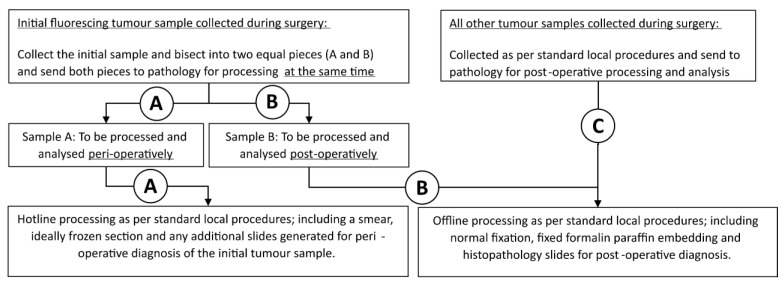
Tumour sample collection and processing. A = peri-operative processing pathway; B = post-operative processing pathway; C = Standard of Care tumour sample collection pathway.

**Figure 3 jpm-13-00514-f003:**
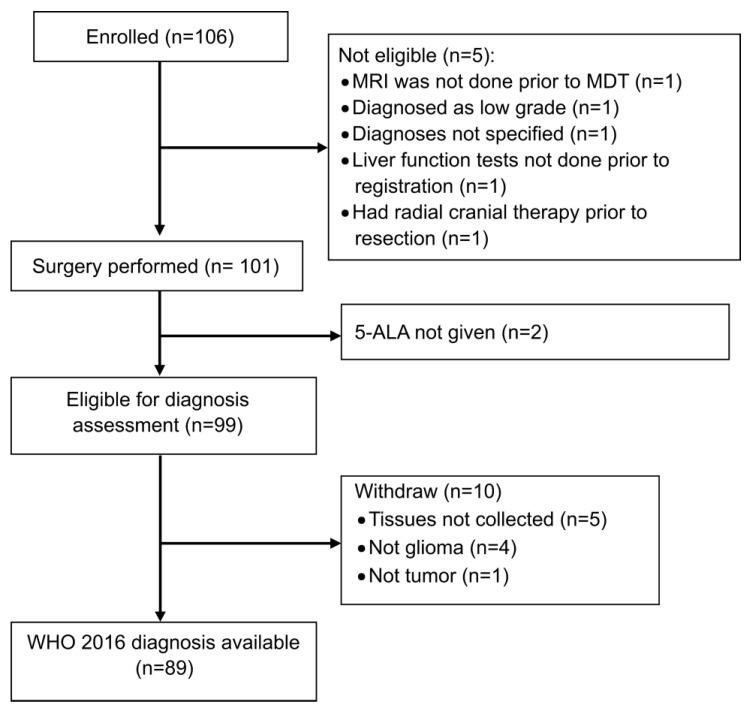
Consort Diagram.

**Figure 4 jpm-13-00514-f004:**
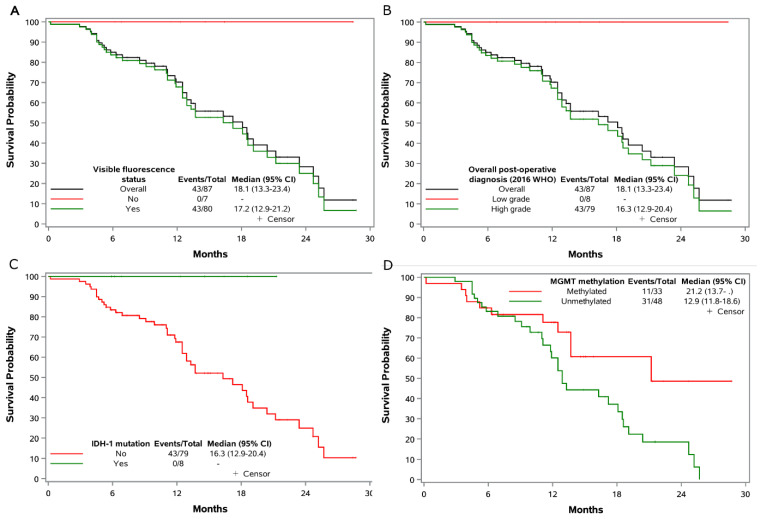
Overall survival in the eligible patients. Shown are Kaplan–Meier curves for overall survival. Tick marks indicate censored data. The survival time calculate from date of registration to date of death, censored if alive or lost to follow-up. (**A**) Overall survival of overall population and stratified by visible fluorescence; (**B**) Overall survival of overall population and stratified by post-operative diagnosis (2016 WHO); (**C**) Overall survival stratified by IDH-1 mutation status; (**D**) Overall survival stratified by MGMT methylation status. Two patients optout from study, the MGMT methylation status was not available for 6 patients.

**Table 1 jpm-13-00514-t001:** Baseline Characteristics and Surgical Details of patients recruited.

Characteristics		Patients (N = 106)
Age, n/N (%)		
	≥20 and <30	2/106 (1.9)
	≥30 and <40	11/106 (10.4)
	≥40 and <50	12/106 (11.3)
	≥50 and <60	29/106 (27.4)
	≥60 and <70	35/106 (33)
	≥70 and <80	17/106 (16)
Sex, n/N (%)		
	Male	63/106 (59.4)
	Female	43/106 (40.6)
WHO performance status		
	0	54/105 (51.4)
	1	51/105 (48.6)
No. of tumour locations		
	1	101/106 (95.3)
	2	5/106 (4.7)
Tumour details		
	Front	40/111 (36.0)
	Temporal	40/111 (36.0)
	Parietal	20/111 (18.0)
	Occipital	5/111 (4.5)
	Multifocal	6/111 (5.4)
Tumour hemisphere		
	Left	53/106 (50.0)
	Right	53/106 (50.0)
WHO grade evaluated by local radiologist		
	WHO grade II	1/106 (0.9)
	WHO grade IV	84/106 (79.2)
	WHO grade II transforming to a higher grade	20/106 (18.9)
	Other (Not specified)	1/106 (0.9)
Surgery performed, n/N (%)		101/106 (95.3%)
5-ALA taken, n/N (%)		99/101 (98.0)
5-ALA dosing, mg (N = 99)		1500 (960, 2200)
Visible fluorescence of tumour confirmed, n/N (%)		85/99 (85.9)
Considered completed resection by surgeon		75/99 (75.8)
Complete resection of enhancing tumour, n/N (%)		30/63(47.6)
Percentage validation, n/N (%)		
	<90%	10/30 (33.3)
	>90%	20/30 (66.7)

**Table 2 jpm-13-00514-t002:** Summary of local perioperative and central formal diagnosis in patients with the probable high-grade disease and suspected transformation to high-grade disease stratified by visible fluorescence.

Local MRI	Visible Fluorescence	Diagnosis	Local Perioperative	Overall Postoperative (WHO 2016)
Probable high-grade disease on preoperative imaging	No (3/99)	Low grade	3	2
		Missing		1 ^a^
	Yes (78/99)	High grade	62	74
		Low grade		1
		Other	7	
		Unable to report	7	
		Missing	2	3 ^b^
Suspected transformation to high-grade tumour on preoperative imaging	No (11/99)	Low grade	7	6
		Other	1	
		Unable to report	2	
		Missing	1	5 ^c^
	Yes (7/99)	High grade	4	6
		Other	1	
		Unable to report	2	
		Missing		1 ^d^

^a^, Patient was not glioma; ^b^, tissues not collected for 2 (1 unable to report and 1 missing in local perioperative diagnosis) patients and the other one is not glioma (local perioperative diagnosis is missing); ^c^, tissues not collected for 2 (1 unable to report and 1 missing in local perioperative diagnosis) patients, 2 was not glioma (1 low grade and 1 unable to report in local perioperative diagnosis) and 1 was not tumour (diagnosed as other in local peri-operative diagnosis); ^d^, tissues not collected (diagnosed as high grade in local perioperative).

**Table 3 jpm-13-00514-t003:** Diagnosis (WHO 2021) of patients by fluorescence status for patients with available diagnosis.

		Visible Fluorescence Confirmed		
Variables		No (N = 8)	Yes (N = 81)	Total (N = 89)
Overall postoperative diagnosis, n/N (%)				
	WHO grade I	1/8 (12.5)	0	1/89 (1.1)
	WHO grade II	7/8 (87.5)	1/81 (1.2)	8/89 (9)
	WHO grade III	0	3/81 (3.7)	3/89 (3.4)
	WHO grade IV	0	77/81 (95.1)	77/89 (86.5)
Overall postoperative diagnosis, n/N (%)				
	Low grade	8/8 (100)	1/81 (1.2)	9/89 (10.1)
	High grade	0	80/81 (98.8)	80/89 (89.9)

## Data Availability

The trial data is held by the Cambridge Clinical Trials Unit-Cancer Theme. Ownership of the data resides with the Trial Management Group (TMG). Access to the data can be requested and authorsed by the TMG.

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
