# Peer review of "Refining the Intraoperative Identification of Suspected High-Grade Glioma Using a Surgical Fluorescence Biomarker: GALA BIDD Study Report"

_jpm, 2023, doi:10.3390/jpm13030514_

Round 1
Reviewer 1 Report
Watts et al. studied on the topic of “Improving the intra-operative identification of suspected high- 2 grade glioma using a surgical fluorescence biomarker.” The authors described their work in an efficient way and this manuscript falls within the scope of a journal like JPM. However, I pointed out some minor flaws that should be rectified before publication.
1. A more detailed description of how 5-aminolevulinic acid (5-ALA) exhibits fluorescence is required in the introduction part also cites some current references. (Major)
2. The manuscript mostly projected on 5-ALA as a fluorescence biomarker. However, the article is lacking a detailed description of accrued fluorescence. Also, a clear description is needed for fluorescence mapping and intensity variation. Otherwise, it looks like the fluorescence behavior of 5-ALA is solely dependent upon visibility. Then, what is the visibility parameter? Does this exclude human biases? (Major)
3. Some AND and NOT are represented as uppercase. Is this have any purpose? (Minor)
4. Previous revision corrections are visible throughout the manuscript. Remove those. (Minor)
5. The image quality of the tables is not good. Replace with new one. (Minor)
6. Page no 3, line number 253 what is the significance of a lot of asterisk signs? (Minor)
7. Check the reference style and abbreviation of journals. (Major)
8. Some typos are there; rectify those. (Minor)
Reviewer 2 Report
Watts et al describe the usage of the surgical fluorescent biomarker 5-ALA during of high grade glioma in the GALA BIDD study. The study is well conducted and the manuscript is well-written, but a little bit of concerns remains for the publication of the present form.
Major:
1. About the title, what is “improving”? The 5-ALA is a useful fluorescent marker during the operation has already known for 20 years.
2. Comparisons with other fluorescent biomarkers, such as talaporfin, may also benefit in the Discussion.
Minor:
3. In the structural abstract, bold font for “Background”, “Methods”, “Results”, and “Conclusion” is better.
Round 2
Reviewer 2 Report
The authors answer appropriately to the reviewer's comments.
Author Response
Thank you for your very helpful comments:
In response, we have updated the manuscript to be in line with the latest WHO 2021 classification Lines 296-311 (grade 3 astrocytoma NOS) as requested.
We had outlined our strategy below: Lines 182- 189
Since the study was completed the WHO 2021 classification has been published, in which the grading incorporates molecular and histological parameters For the purposes of our study the main histological grading criteria which are used at the time of intra-operative diagnosis have not changed in the 2021 classification and therefore we have opted to use the classification given at the time where possible incorporating the molecular information that was available at the time of the study.